# Comparison of mRNA Vaccinations with BNT162b2 or mRNA-1273 in Anti-CD20-Treated Multiple Sclerosis Patients

**DOI:** 10.3390/vaccines10060922

**Published:** 2022-06-09

**Authors:** Helly Hammer, Robert Hoepner, Christoph Friedli, Stephen L. Leib, Franziska Suter-Riniker, Lara Diem, Nicole Kamber, Andrew Chan, Anke Salmen, Christian P. Kamm

**Affiliations:** 1Department of Neurology, Inselspital, Bern University Hospital and University of Bern, 3010 Bern, Switzerland; robert.hoepner@insel.ch (R.H.); christoph.friedli@insel.ch (C.F.); larafrancesca.diem@insel.ch (L.D.); nicole.kamber@insel.ch (N.K.); andrew.chan@insel.ch (A.C.); anke.salmen@insel.ch (A.S.); christian.kamm@luks.ch (C.P.K.); 2Institute for Infectious Diseases, University of Bern, 3010 Bern, Switzerland; stephen.leib@unibe.ch (S.L.L.); franziska.suter@unibe.ch (F.S.-R.); 3Neurocentre, Luzerner Kantonsspital, 6016 Lucerne, Switzerland

**Keywords:** anti-CD20, COVID-19, multiple sclerosis, pandemic, SARS-CoV-2, vaccination

## Abstract

*Objective*: Anti-CD20-treated patients are at risk of a reduced humoral immune response during the SARS-CoV-2 pandemic. Our aim was to compare the antibody response after two vaccinations with the mRNA vaccines BNT162b2 or mRNA-1273 in patients with multiple sclerosis. *Methods*: Data from the University Hospital of Bern and Cantonal Hospital of Lucerne were retrospectively collected from medical records and then analyzed. Anti-spike IgG serum titers were collected from both centers and were considered to be protective from a value of ≥100 AU/mL. Continuous variables were given as the mean and 95% confidence interval (95% CI); categorical variables were given as frequencies. A Mann–Whitney test and Fisher’s exact test as well as a multivariable linear regression analysis with anti-spike IgG (AU/mL) as the dependent variable were run using SPSS Statistic 25 (IBM Corp., Amonk, NY, USA). *Results*: A total of 74 patients were included; 41/74 (63.51%) were female patients and the mean age was 46.6 years (95% CI 43.4–49.9). Of these patients, 36/74 were vaccinated with BNT162b2 and 38/74 with mRNA-1273, following the national vaccination recommendation. In both vaccine groups, protective anti-spike IgG titers (≥100 AU/mL) were infrequently achieved (5/74: mRNA-1273 3/38; BNT162b2 2/36). *Conclusions*: In addition to a low rate of protective anti-spike IgG titers in both vaccine groups, we identified a drop in anti-spike IgG serum titers over time. This observation bears therapeutic consequences, as initial positive titers should be checked in case of an infection with the SARS-CoV-2 virus to identify patients who would benefit from an intravenous anti-spike IgG treatment against acute COVID-19.

## 1. Introduction

The severe acute respiratory syndrome coronavirus type 2 (SARS-CoV-2) pandemic, which began at the end of 2019, created new challenges in different domains; this was especially the case for health care systems and their physicians as well as patients with immunosuppressant therapies. To date, the pandemic remains challenging, particularly regarding vaccinations and vaccination counseling for patients at risk. These include elderly patients, patients with comorbidities (e.g., diabetes, obesity and hypertension) or patients treated with immunotherapy or disease-modifying therapies (DMT). Anti-CD20 therapies are used in the treatment of, for example, rheumatological diseases, hematological malignancies or neurological disorders such as multiple sclerosis (MS) [1,2]. The aim of the therapy in patients with multiple sclerosis is to reduce disease activity [2]. B cell depletion occurs up to 72 h hours after infusion and lasts up to 9 months and in some cases even longer. B cell repopulation can usually be expected after 9–12 months [3,4].

Until now, vaccination strategies have varied from country to country. Anti-CD20 treatments have repeatedly been shown to be associated with a decreased humoral immune response after a vaccination [2,5]. The superiority of the vaccination effects in patients treated with anti-CD20 treatments of either of the available mRNA vaccines, Moderna^®^ (mRNA-1273, Cambridge, MA, USA) and Pfizer/BioNTech^®^ (BNT162b2, Mainz, Rhineland-Palatinate, Germany), has not yet been demonstrated. A cohort study, including untreated multiple sclerosis (MS) patients and MS patients treated with several different disease-modifying therapies (DMTs), demonstrated in the total population by a multivariable analysis a higher antibody titer in patients vaccinated with mRNA-1273 compared with BNT162b2 [6].

As anti-CD20-treated patients are particularly at risk of a reduced humoral response to mRNA vaccinations as well as a more severe course of COVID-19, our cohort study aimed to compare the antibody response after two vaccinations with either BNT162b2 or mRNA-1273 in MS patients treated with anti-CD20 therapies.

## 2. Materials and Methods

Anti-CD20-treated MS patients who had been vaccinated twice with either BNT162b2 or mRNA-1273 as indicated by SwissMedic were retrospectively identified by a medical chart review. Of the 74 included patients, 26 were identified at Bern University Hospital (ethical vote of NI registry study: 2017-01369, last amendment August 2020) and 48 were identified at the Cantonal Hospital of Lucerne (ethical vote: 2020-00044). Anti-spike IgG serum titers were assessed using an Abbott SARS-CoV-2 IgG assay (Abbott Laboratories, Chicago, IL, USA) and a Liaison SARS-CoV-2 S1/S2 IgG assay (Diasorin, Italy) in Bern and an Elecys^®^ SARS-CoV-2 Ig assay (Roche, Switzerland) in Lucerne. According the manufacturer’s instructions, the Liason SARS-CoV-2 S1/S2 IgG assay was evaluated with a plaque reduction neutralization test (PRNT); 87% of the samples with a SARS-CoV-2 S1/S2 IgG result ≥ 80 AU/mL had a PRNT titer of ≥1:160. Therefore, we proposed a cut-off of 100 AU/mL and assumed at both centers an anti-spike titer of ≥100 AU/mL to be protective. B cell counts were analyzed in the main laboratory of each center (Bern and Lucerne, Switzerland) using fluorescence-activated cell sorting (FACS). The continuous variables were given as a mean and a 95% confidence interval (95% CI); the categorical variables were given as frequencies. A Mann–Whitney test and a Fisher’s exact test as well as a multivariable linear regression analysis with anti-spike IgG (AU/mL) as the dependent variable were run using SPSS Statistic 25 (IBM Corp., Amonk, NY, USA).

## 3. Results

Of the 74 patients, 41 (63.51%) were female, the mean age was 46.6 years (95% CI 43.4–49.9) and a diagnosis of relapsing multiple sclerosis (RMS) was more frequent (57/74, 77.02%) than primary progressive MS (PPMS) (17/74, 23.61%). A total of 9/74 were treated with Rituximab (MabThera^®^, Roche) and 65/74 with Ocrelizumab (Ocrevus^®^, Roche). In total, 36/74 were vaccinated with BNT162b2 and 38/74 with mRNA-1273, following the national vaccination recommendations (mRNA-1273: 2 vaccinations (100 μg/0.5 mL per vaccination) with an interval of 28 days; BNT162b2: 2 vaccinations (30 μg/0.3 mL per vaccination) with an interval of 21 days; Appendix A) [7]. In both vaccine groups, protective anti-spike IgG titers (≥100 AU/mL) were infrequently achieved (5/74: mRNA-1273 3/38; BNT162b2 2/36) (Appendix A). The anti-spike IgG titer did not differ between the two vaccine types. In order to include the distribution of cohort characteristics in our analysis, we ran a multivariable linear regression analysis. This demonstrated no significant differences between both vaccines (Figure 1; Table 1). However, in this regression analysis, the time interval between the measurement of the humoral immune response and the second vaccination demonstrated a trend toward a reduction in anti-spike IgG serum titers over time in anti-CD20-treated MS patients (Table 1). Furthermore, in patients treated with anti-CD20 medications, data on B cell counts were present in 68 cases. The mean percentage of B cells was 0.68% (95% CI 0.16–1.09). Overall, the anti-spike IgG titer showed a non-significant trend toward lower values in patients with a complete B cell depletion (0.0%, spike titer mean 95% CI 16.69; 5.9–39, 3) compared with those with an incomplete B cell depletion (>0.0%, spike titer 51.9; 6.58–97.22, Mann–Whitney test *p* = 0.14), which became significant when comparing those with a protective anti-spike IgG titer level with patients without a protective antibody response (Chi2 *p* = 0.04; Appendix A).

## 4. Discussion

In both vaccine groups, protective anti-spike IgG titers (≥100 AU/mL) were infrequently achieved. This was reasonable due to the mode of action of B cell-depleting drugs that destroy CD20+ B cells and, therefore, suppress antibody production—for example, after a vaccination. In greater detail, the anti-spike IgG responses between the two mRNA vaccines (BNTech-162b2 vs. mRNA-1273) did not differ in our cohort, either regarding the titer levels or regarding reaching the threshold for a protective humoral response. We observed a trend that anti-CD20-treated patients showed a drop in anti-spike IgG serum titers over time. This observation, which has also been shown by others [6], bears therapeutic consequences as initial positive titers should be checked in case of an infection with the SARS-CoV-2 virus to identify patients who would benefit from an intravenous anti-spike IgG treatment against acute COVID-19. However, it has to be noted that it is unknown to what extent a drop of anti-spike IgG serum titer over time corresponds with a diminished immune response to COVID-19, which has to be shown in larger clinical trials. A reduced protection against the virus can, however, be anticipated by advocating booster vaccinations in these patients. The main limitation of our work in addition to the retrospective design was the small sample size, which affected the generalizability and power of our analysis. Furthermore, as this was a real-world study, different assays were used to detect the SARS-CoV-2 immune response run in the main laboratories of the respective centers. The assay used in Bern detected IgG whereas the assay run in Lucerne detected high-affinity Ig against the anti-spike region, which mainly consists of IgG but smaller amounts of IgM antibodies are also measured. To statistically adjust for this methodological shortcoming of this retrospective real-world study, the multivariable linear regression analysis was controlled for each center (Bern vs. Lucerne), which did not impact the anti-spike IgG level. The worldwide pandemic remains a major threat for our societies, calling for open data and data-sharing strategies to answer the major questions as a research community (see Data Availability Statement below).

## 5. Conclusions

We identified a trend that anti-CD20-treated patients showed a drop in anti-spike IgG serum titers over time despite the same anti-spike IgG responses between the two mRNA vaccines. This observation, which has also been shown by others, bears therapeutic consequences as initial positive titers should be checked in case of an infection with the SARS-CoV-2 virus to identify patients who would benefit from an intravenous anti-spike IgG treatment against acute COVID-19.

## Figures and Tables

**Figure 1 vaccines-10-00922-f001:**
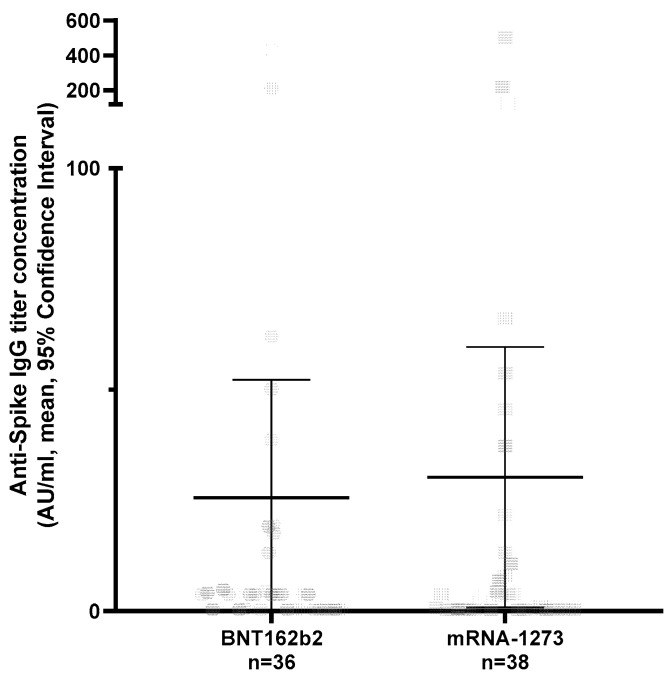
Anti-Spike IgG titer in anti-CD20 treated MS patients vaccinated twice, following the SwissMedic recommendations, with BNT162b2 or mRNA-1273 vaccine. Serum was used for analysis. Abbreviations: AU: arbitrary units, ml: milliliter.

**Table 1 vaccines-10-00922-t001:** Multivariable linear regression analysis with the dependent variable of anti-spike IgG (AU/mL) in anti-CD20-treated MS patients. Statistics: collinearity of each variance inflation factor (VIF) < 1.5, R^2^ = 0.11, model (*p*) = 0.45. CI: confidence interval; LL: lower limit; Ref: reference; RMS: relapsing multiple sclerosis; UL: upper limit.

Variables	Coefficient	95% Confidence Interval		*p*-Value
		LL	UL	
Age (years)	−0.37	−1.91	1.17	0.63
Female sex (male sex Ref.)	−32.33	−73.07	8.41	0.12
Lucerne center (Bern Ref.)	19.89	−29.34	69.13	0.42
Diagnosis of PPMS (RMS Ref.)	−15.02	−41.40	11.36	0.26
Anti-CD20 treatment with Rituximab (Ocrelizumab Ref.)	21.99	−50.43	94.42	0.55
Time between vaccination and sampling (years)	−150.43	−309.21	8.35	0.06
Time between last dosage and vaccination (years)	3.03	−66.35	72.41	0.93
mRNA-1273 (BNT162b2 Ref.)	−4.72	−49.32	39.88	0.83

## Data Availability

In terms of the further use of our data, we ask researchers to cite our paper in their Method section. Anonymized data are available via the corresponding author.

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
