# Peer review of "Comparison of mRNA Vaccinations with BNT162b2 or mRNA-1273 in Anti-CD20-Treated Multiple Sclerosis Patients"

_vaccines, 2022, doi:10.3390/vaccines10060922_

Round 1

Reviewer 1 Report

This brief report “Comparison of mRNA Vaccinations with BNT162b2 or mRNA2 1273 in Anti-CD20 Treated Multiple Sclerosis Patients” analyses the effect of two vaccinations with BNT162b2 or MRNA_1273 mRNA vaccines in anti-CD20-treated MS patients to assess the humoral immune response to SARS-CoV2.

Early monitoring of immunoglobulin levels might help to identify the risk for developing infection in anti-CD20-treated MS patients.

There are some aspects that should be improved before its publication:

  • The limitation of the study is the small number of patients treated with fingolimod. I suggest that the number of patients treated with fingolimod should be increased.
  • B cell depletion therapies targeting CD20 deplete all B cells except antibodies secreting cells and pro-B cells and have shown significant efficacy. In rituximab-treated patients, a reduction of B cells was observed in RRMS while PPMS were only a moderate reduction. In my opinion, the % of B-lymphocytes should be assessed.
  • As a complementary data, I believe that the IgM value should be added.

In conclusion, the topic and objectives of the paper are interesting and important for the field of multiple sclerosis.

Reviewer 2 Report

In the manuscript of Hammer and al reports  evaluation of mRNA vaccines NT162b2 or mRNA-1273 immune respose efficiency in anti-CD-20 treated patients.

The report is clearly written. Overall, this study confirms the previous observations regarding the drop of IgG titers for both vaccine groups but it is important findings that response on vaccination was very low in in anti-CD-20 treated patients. Based on publish data mRNA-1273 vaccine elicits a higher IgG titer in non-immunocompromised persons, but this study demonstrated no difference between both vaccine groups.

The study presents valuable data which may contribute to the COVID-19 treatment.

Author Response

Thank you for your comments. In view of this, we have not made any changes.

Reviewer 3 Report

Critiques:

1. Please explain how the 100AU was selected as a cutoff for response to COVID vaccine.

2. The Figure should be part of the manuscript, not a supplementary material.

3. Discuss why protective anti-spike IgG titers (≥100 22 AU/ml) were infrequently achieved in patients treated with anti-Cd20 therapies.
